# Predictors of Hepatitis B screening and vaccination status of young psychoactive substance users in informal settlements in Kampala, Uganda

Tonny Ssekamatte[1]*, John Bosco Isunju[1], Joan Nankya Mutyoba[2], Moses Tetui[3], Richard K. Mugambe[1], Aisha Nalugya[1], Winnifred K. Kansiime[1], Chenai Kitchen[4], Wagaba Brenda[1], Patience Oputan[1], Justine Nnakate Bukenya[5], Esther Buregyeya[1], Simon P. S. Kibira[5]

1 Department of Disease Control and Environmental Health, Makerere University School of Public Health, Kampala, Uganda, 2 Department of Epidemiology and Biostatistics, Makerere University School of Public Health, Kampala, Uganda, 3 Department of Health Policy Planning and Management, Makerere University School of Public Health, Kampala, Uganda, 4 Department of Pharmacy Administration and Clinical Pharmacy, School of Pharmacy, Xi'an Jiaotong University, Xi'an, China, 5 Department of Community Health and Behavioural Sciences, Makerere University School of Public Health, Kampala, Uganda

* ssekamattet.toca@gmail.com, tssekamatte@musph.ac.ug

## Abstract

### Background

Young psychoactive substance users exhibit high-risk behaviours such as unprotected sexual intercourse, and sharing needles and syringes, which increases their risk of Hepatitis B infection. However, there is limited evidence of screening, and vaccination status of this subgroup. The aim of this study was to establish the predictors of screening and completion of the hepatitis B vaccination schedule.

### Methods

A cross-sectional study using respondent driven sampling was used to enrol respondents from twelve out of fifty-seven informal settlements in Kampala city. Data were collected using an electronic structured questionnaire uploaded on the KoboCollect mobile application, and analysed using Stata version 14. A "modified" Poisson regression analysis was done to determine the predictors of screening while logistic regression was used to determine the predictors of completion of the Hepatitis B vaccination schedule.

### Results

About 13.3% (102/768) and 2.7% (21/768) of the respondents had ever screened for Hepatitis B, and completed the Hepatitis B vaccination schedule respectively. Being female (aPR 1.61, 95% CI: 1.11–2.33), earning a monthly income >USD 136 (aPR 1.78, 95% CI: 1.11–2.86); completion of the Hepatitis B vaccination schedule (aPR 1.85, 95% CI: 1.26–2.70); lack of awareness about the recommended Hepatitis B vaccine dose (aPR 0.43, 95% CI:

**Data Availability Statement:** All relevant data are within the paper and its Supporting information files.

**Funding:** The author(s) received no specific funding for this work.

**Competing interests:** The authors have declared that no competing interests exist.

**Abbreviations:** HBV, Hepatitis B virus; RDS, Respondent Driven Sampling; UBOS, Uganda Bureau of Statistics; WHO, World Health Organisation.

0.27–0.68); and the belief that the Hepatitis B vaccine is effective in preventing Hepatitis B infection (aPRR 3.67, 95% CI: 2.34–5.73) were associated with "ever screening" for Hepatitis B. Knowledge of the recommended Hepatitis B vaccine dose (aOR 0.06, 95% CI: 0.01–0.35); "ever screening" for hepatitis B (aOR 9.68, 95% CI: 2.17–43.16) and the belief that the hepatitis B vaccine is effective in preventing Hepatitis B infection (aOR 11.8, 95% CI: 1.13–110.14) were associated with completion of the hepatitis B vaccination schedule.

## Conclusions

Our findings indicate a low prevalence of Hepatitis B screening and completion of the Hepatitis B vaccination schedule among young psychoactive substance users in informal settings. It is evident that lack of awareness about Hepatitis B is associated with the low screening and vaccination rates. We recommend creation of awareness of Hepatitis B among young people in urban informal settlements.

## Background

The use of psychoactive substances such as alcohol, marijuana (*Cannabis sativa*), amphetamines, oral tobacco, heroin and khat (*Catha edulis)* remains a significant global public health challenge [1]. Psychoactive substance use is associated with having multiple sexual partnerships, unprotected sexual intercourse, at times with individuals whose health status may not be known, drug–sex exchanges and sharing of drug preparation equipment such as used needles and syringes [2–4]. Engaging in these behaviours increases the risk of transmission of Hepatitis B virus (HBV) infection [2, 5–7]. Hepatitis B is a life-threatening infection caused by HBV [8]. In 2019, HBV accounted for over 820,000 deaths globally [9], among which 1,206 were reported in Uganda [10]. HBV infection is highly endemic in Uganda, with a seroprevalence of 4.3% (5.6% among men and 3.1% among women) and a lifetime exposure of the population as high as 52% [11].

Owing to the public health significance of Hepatitis B [12, 13], the World Health Organisation (WHO) developed guidelines on Hepatitis B testing in which it emphasized the need for testing high-risk subgroups such as psychoactive substance users [14]. These guidelines aimed to strengthen and expand Hepatitis B testing/screening, vaccination and linkage into care [14, 15]. Screening and vaccination are the cornerstone of Hepatitis B prevention [16]. Screening presents an opportunity for health education, counselling on risky practices and provision of sterile needles to injecting substance users [15], while vaccination can reduce the incidence of HBV infection [3, 9].

WHO recommends that psychoactive substance users including injection drug users should be tested for and vaccinated with three doses of the Hepatitis B vaccine [8]. Despite these recommendations, Hepatitis B screening and vaccination programs targeting psychoactive substance users are still uncommon especially in informal settlements [17]. Even where services are available, psychoactive substance users often shy away from these programs due to the costs involved and fear to commit to the vaccination schedule [17, 18]. Consequently, only a small percentage of individuals with Hepatitis B either know their serostatus or are able to access appropriate treatment and care [19]. The majority often report with the advanced disease [14].

In Uganda, the Ministry of Health (MoH), in 2002, incorporated early childhood vaccination against hepatitis B (given at the age of 6, 10 and 14 weeks) into its expanded program on

immunization [20]. Besides early childhood vaccination, the MoH promotes injection safety, screening donor transfusions for blood borne infections, and vaccination of high-risk groups including adolescents and substance users [20, 21]. Despite these interventions, a large cohort of people, including psychoactive substance users are still being infected with HBV [11, 20]. HBV infection is associated with hepatocellular carcinoma, liver cirrhosis, inflammation and early death [14, 22]. These outcomes pose a serious economic burden on both the healthcare system and households [23–25]. Besides, evidence on screening and vaccination against hepatitis B among substance users in informal settlements in low-and-middle income countries like Uganda is limited.

Evidence on the predictors of Hepatitis B screening and vaccination status, will inform interventions aimed at achieving the targets highlighted in the global hepatitis response strategy [15]. Our study used the Andersen's behavioural framework of health care utilization [26–28] and the knowledge, attitude and practice model [29–31] to establish the predictors of Hepatitis B screening and vaccination status of young psychoactive substance users in the informal settlements of Kampala, Uganda. These models have previously been applied to understand Hepatitis B screening and vaccination behaviours [32–34].

## Materials and methods

### Scope and design

A cross-sectional survey was conducted between June and July 2019 in the informal settlements of Kampala, Uganda's largest urban centre and capital. Data were collected between 8:00 am and 6:00 pm. Kampala's population is estimated at 1.5 million people, 27.5% of whom are aged between 15–24 years [35]. The city has five metropolitan administrative divisions and is home to Uganda's national referral hospitals. Data analysed for the current sub study were collected as part of a larger study titled "High-risk sexual behaviours of young psychoactive substance users in Kampala's informal settlements, Uganda" [36]. We defined an informal settlement as an urban residence characterised by inadequate access to social services and poor structural quality of housing, overcrowding and insecure residential status [37].

### Study population and eligibility criteria

In order to be eligible to participate in the current study, a respondent must have been aged 18–24 years or age, and a current user of at least alcohol, heroin, marijuana, khat or oral tobacco. Users of alcohol, heroin, marijuana, khat and oral tobacco were studied since the use of these substances is associated with unsafe sexual practices such as inconsistent condom use [38–40]. Unsafe practices such as unprotected sexual intercourse with an infected person are known to increase the risk of hepatitis B infection [41–44]. Young people aged 18–24 years are above the legal age, meaning that they have the autonomy to engage in psychoactive substance use and high-risk sexual encounters without parental restrictions or consent [45], which increases risk of sexual transmission of HBV infection. In addition, a respondent must have stayed in the selected informal settlement for a period not less than 6 months, so as to make it easy for the peers (these were used in the recruitment of the respondents) to ascertain whether they resided in the study area and if they were substance users or not. During the recruitment of the study participants, both the primary and secondary seeds, were briefed on the inclusion and exclusion criteria. This reduced the probability of seeds enrolling peers who were not eligible to participate, and consequently non-response. Respondents who were sick or under the influence of a psychoactive substance were not enrolled for the study.

## Sample size and sampling procedures

The sample size was calculated using the Kish Leslie formula for cross-sectional studies [46]. Since there was limited evidence on the prevalence of Hepatitis B screening or completion of the Hepatitis B vaccination schedule among young psychoactive substance users in informal settlements, we chose a conservative prevalence of 50% [47], a 95% level of confidence, a margin of error (d) of 0.05 and a design effect of 2.0 [47, 48] so as to determine the sample size. This yielded a sample size of 768 young psychoactive substance users. A total of 12 out of 57 informal settlements [49], were purposively selected for geographical representation of the informal settlements in the city. The informal settlements included in the study have been reported in our previous publications [16, 36].

After the purposive selection of informal settlements, respondent driven sampling was used in the selection of study participants. For each of the informal settlements, we used community leaders who had participated in a previous study in the informal settlements to identify four individuals who acted as primary seeds [50]. During enrolment of primary seeds, research assistants made sure that the selected individuals were not under the influence of psychoactive substances. The selected seeds were first interviewed by research assistants prior to being given coupons to enrol secondary seeds. The secondary seeds were then requested by the primary seeds to report at an agreed venue where they were screened for eligibility using a checklist prior to providing informed consent and consequently interview. The details of our sampling methodology have been published in our earlier studies [16, 36].

## Variable measurement

The main outcomes of interest in this study were having undergone hepatitis B screening in the last 12 months, and completion of the Hepatitis B vaccination schedule based on the WHO recommendation of 3 vaccine doses [51]. However, the completion was irrespective of the timing of the vaccinations. We defined vaccine uptake as the number of people vaccinated with a certain dose of the vaccine in a certain time period, expressed as the proportion of a target population [52]. Self-reports were used to measure completion of the vaccination schedule. The independent variables considered in this study were informed by the Andersen's behavioural framework of health care utilization [27] and the knowledge, attitude and practice (KAP) model [53]. According to the Andersen's behavioural framework, utilisation of healthcare services such as screening and vaccination against hepatitis B is influenced by the environment in which the individual lives, and population characteristics and health behaviours [27]. The KAP model suggests that the uptake of healthcare services such as screening and vaccination is influenced by individuals' level of knowledge, attitude and risk perception.

Based on the Andersen's behavioural framework of health care utilization, the KAP model and a review of literature, the independent variables of interest in this study were socio-demographic characteristics such as sex, age, level of education, and knowledge, risk perception and attitude towards Hepatitis B screening and vaccination. History of substance use was classified as "ever used' which referred to lifetime use of a psychoactive substance; 'recent use' which referred to having used a psychoactive substance in the last 12 months and 'current use' referring to the use of a psychoactive substance in the last 30 days. Knowledge was assessed using questions on the recommended Hepatitis B vaccine dose and the duration the vaccine protects someone against the Hepatitis B infection. Attitude was measured using a question on the perceived efficacy of Hepatitis B vaccine. Age, marital status, level of education, income levels, risk perception, attitude, ever screening for hepatitis B vaccination status have been shown by previous scholars to influence ether an individual's screening status or completion of vaccination schedules [16, 54–60].

## Data collection tool and quality control measures

A structured questionnaire was designed using the kobo tool box online platform, and later uploaded onto the KoboCollect mobile application. The KoboCollect application was pre-installed on smart phones and tablets. The structured questionnaire was designed with skips to reduce errors by research assistants. The questions used in the current study were adopted from the data collection tool that was used by Ssekamatte, Mukama [16] to establish the screening and vaccination status of healthcare providers in Wakiso district, Uganda. The data collection tool was validated by a team of experts in hepatitis B research who were based at the Makerere University College of Health Sciences [31, 60]. Prior to data collection, all research assistants received training on the study protocol and data collection tool. The pre-test enabled the research assistants to familiarise with the data collection tool and the psychoactive substance users' community. All study tools were translated into the local language (Luganda) and thereafter pretested. The data collection tool was pretested among 20 young psychoactive substance users in an informal settlement in Kajjansi Town Council, Wakiso district, and relevant adjustments were made. An informal settlement in Kajjansi Town council was chosen since it had characteristics similar to those of Kampala's informal settlements. In particular, the settlement had a high population of young psychoactive substance users. It was apparent that some participants could be influenced by one or more psychoactive substances, which would alter their emotional state, perception, judgement and performance [61]. We therefore trained research assistants to be sensitive to behavioural signs of intoxication such as loss of co-ordination, staggering gait, drowsiness, slurred speech and glazed eyes for alcohol users, and paranoia, anxiety, eye-rolling, pupil dilation/constriction, head movements or jerks for other substances [61]. Research assistants were also cautioned to remind the participants of their right to withdraw, particularly when observable signs of intoxication appeared to change [61]. This enabled the data collection team to obtain data from participants who were not intoxicated thereby improving the quality of the data.

## Data management and analysis

Data were collected using smart phones and tablets, and later uploaded to an online server at; https://kobo.humanitarianresponse.info. Upon submission, the data were reviewed on a daily basis by the principal investigators for consistency. Prior to analysis, data were downloaded in a Microsoft Excel format and further cleaned to reduce any possible errors. Measures of central tendency such as means, median and mode were particularly used to identify errors in the continuous variables. Data were analysed using STATA version 14.0. Descriptive statistics were performed to summarize both continuous and categorical variables (background characteristics of respondents, history of substance use, prevalence of Hepatitis B screening and completion of the Hepatitis B vaccination schedule).

Inferential statistics were used to determine the predictors of Hepatitis B screening and completion of the Hepatitis B vaccination schedule. A modified Poisson regression analysis was used to determine the predictors of Hepatitis B screening since the prevalence of screening for Hepatitis B was greater than 10% [62, 63]. Bivariate analysis was done first to establish the association between predictor variables and screening for Hepatitis B. A cut off p-value of less than 0.2 was set for variables eligible to be included in the multivariable model [64]. Prevalence ratios (PR) and their corresponding 95% confidence intervals were used as the measure of risk.

Given that the prevalence of completion of the Hepatitis B vaccination schedule was a rare occurrence (less than 10%) among young psychoactive substance users, we used logistic regression to establish the predictors. Initially, bivariate logistic regression was used to determine the predictors of completion of the hepatitis B vaccination schedule. Predictors that had

a p-value of less than 0.2 were included in the multivariable model. Odds Ratios (OR) were used as the appropriate measure of risk.

### Ethics approval and consent to participate

The study protocol was approved by Makerere University School of Public Health Higher Degrees and Research Ethics Committee. Given the sensitivity of the study population, permission to interview the study participants was also sought from the local authorities and from peer leaders within the communities where data were collected. Written informed consent was also sought from all study participants prior to participating in the study. The research assistants did not record the names of the respondents on any study forms so as to reduce the risk of breaching confidentiality, and consequently exposing the respondents' identity to the law enforcers such as police, since the use of substances such as marijuana was illicit.

## Results

### Socio-demographic characteristics of respondents

A total of 768 participants were enrolled (response rate of 99.7%). The mean age (SD) of respondents were 21.5±2.1 years. More than three quarters, 78.5% (603/768) of respondents were male, 39.2% (301/768) were Catholic, 78.9% (606/768) had never married, and 64.6% (496/768) reported earning less than USD 68.0 per month (Table 1).

### Psychoactive substance use among young people in Kampala's informal settlements

About 74% (568/768), 54.3% (417/768) and 52% (399/768) were current users of alcohol, khat and marijuana respectively. In addition, 9.2% (71/768) and 1.7% (13/768) were current users of oral tobacco and heroin (Fig 1).

**Table 1. Background characteristics of the psychoactive substance users in Kampala's informal settlements.**

| Characteristic | Category | Frequency (n = 768) | Percentage (%) |
|---|---|---|---|
| Age Mean (SD) = 21.5±2.1) | 18–19 | 190 | 24.7 |
| | 20–24 | 578 | 75.3 |
| Sex | Male | 603 | 78.5 |
| | Female | 165 | 21.5 |
| Marital status | Never married | 606 | 78.9 |
| | Married | 162 | 21.1 |
| Religion | Catholic | 301 | 39.2 |
| | Anglican | 128 | 16.7 |
| | Muslim | 231 | 30.1 |
| | Born again/ Pentecostal | 83 | 10.8 |
| | Other religions | 25 | 3.3 |
| Level of education | Primary | 322 | 41.9 |
| | Secondary and above | 446 | 58.1 |
| Years of staying in area | 0–5 years | 279 | 36.3 |
| | 6–10 years | 149 | 19.4 |
| | > 10 years | 340 | 44.3 |
| Average monthly income (in USD) Exchange rate (1 USD = UGX 3676) | ≤ 68.0 | 496 | 64.6 |
| | 68.1–136 | 207 | 27.0 |
| | Above 136 | 65 | 8.5 |

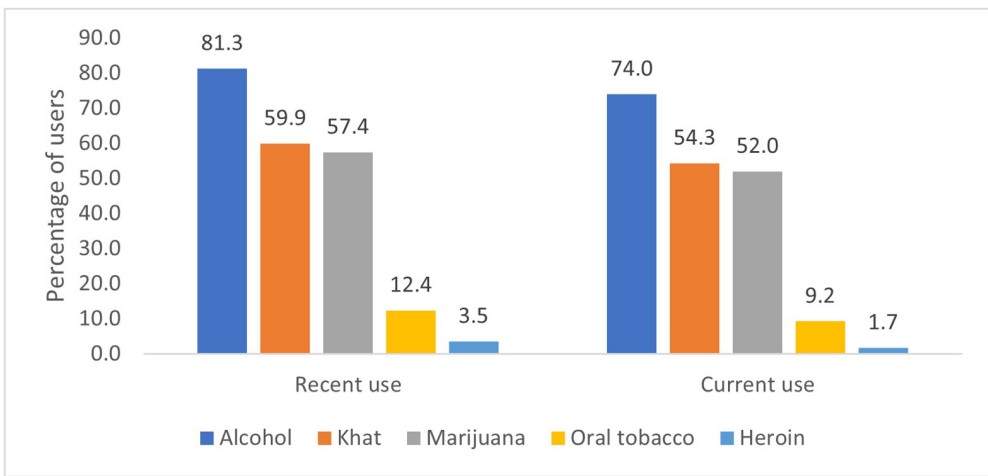

**Fig 1. History of psychoactive substance use among young people in Kampala's informal settlements.**

Hepatitis B screening and vaccination status Only13.3% (102/768) reported ever being screened for HBV infection, among those 5.9% (6/102) reported to have tested positive. Two per cent (16/768) reported having been diagnosed with Hepatitis B in the last 12 months. About 8.0% (62/768) had ever received at least a dose of the hepatitis B vaccine while 2.7% (21/768) had received all the 3 vaccine doses (Fig 2).

## Predictors of Hepatitis B screening

Table 2 shows that sex, level of education, average monthly income, knowledge of the recommended Hepatitis B vaccine doses, Hepatitis B vaccination status and attitude towards the effectiveness of the Hepatitis B vaccine were significantly associated with ever screening for Hepatitis B at multivariable analysis. Females had a 61% higher prevalence of hepatitis B screening compared to males (aPR 1.61, 95% CI: 1.11–2.33, p = 0.01). Young psychoactive substance users who earned more than USD 136.0 had a 78% higher prevalence of hepatitis B screening compared to those who earned less than USD 68.0 (aPR 1.78, 95% CI: 1.11–2.86,

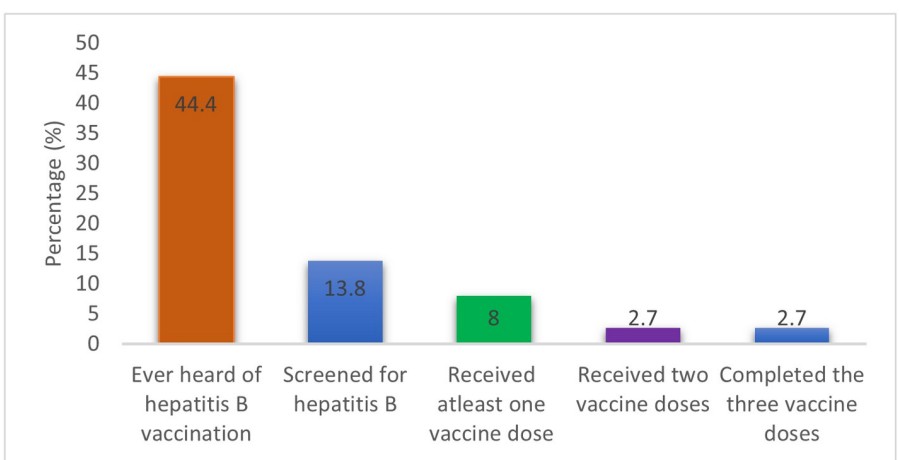

**Fig 2. Hepatitis B testing and vaccination among young psychoactive substance users in Kampala's informal settlements.**

**Table 2. Predictors of "ever screening for hepatitis B" among young psychoactive substance users in informal settlements in Kampala Uganda.**

| Variable | Freq (n) | Ever screened for hepatitis B | | CPR (95% CI) | P-value | aPR (95% CI) | P-value |
|---|---|---|---|---|---|---|---|
| | | Yes | No | | | | |
| **Sex** | | | | | | | |
| Male | 603 | 69 (67.6) | 534 (80.2) | 1 | | | |
| Female | 165 | 33 (32.4) | 132 (19.8) | 1.74 (1.19–2.54) | **0.004** | 1.61 (1.11–2.33) | **0.010*** |
| **Age category** | | | | | | | |
| 18–19 | 190 | 23 (22.5) | 167 (25.1) | 1 | | | |
| 20–24 | 578 | 79 (77.5) | 499 (74.9) | 1.12 (0.73–1.74) | 0.584 | 1.01 (0.67–1.52) | 0.952 |
| **Level of education** | | | | | | | |
| Primary | 322 | 24 (23.5) | 298 (44.7) | 1 | | | |
| Above primary | 446 | 78 (76.6) | 368 (55.3) | 2.34 (1.51–3.62) | **P<.001** | 1.49 (0.99–2.26) | 0.055 |
| **Marital status** | | | | | | | |
| Single | 606 | 77 (75.5) | 529 (79.4) | 1 | | | |
| Married | 162 | 25 (24.5) | 137 (20.6) | 1.21 (0.80–1.84) | 0.361 | | |
| **Still living with parents** | | | | | | | |
| Yes | 120 | 18 (17.6) | 102 (15.3) | 1 | | | |
| No | 648 | 84 (82.4) | 564 (84.7) | 0.86 (0.53–1.38) | 0.543 | | |
| **Average monthly income (USD)** | | | | | | | |
| ≤ 68.0 | 496 | 60 (58.8) | 436 (65.4) | 1 | | | |
| 68.1–136 | 207 | 26 (25.5) | 181 (27.2) | 1.03 (0.67–1.59) | 0.864 | 1.18 (0.79–1.76) | 0.411 |
| Above 136 | 65 | 16 (15.7) | 49 (7.4) | 2.03 (1.24–3.31) | **0.004** | 1.78 (1.11–2.86) | **0.016*** |
| **Know the recommended Hepatitis B vaccine dose** | | | | | | | |
| Yes | 65 | 38 (37.2) | 27 (4.1) | 1 | | | |
| No | 703 | 64 (62.8) | 639 (95.9) | 0.15 (0.11–0.21) | **P<.001** | 0.43 (0.27–0.68) | **<.001*** |
| **Know the duration the vaccine provides protection against HBV** | | | | | | | |
| No | 741 | 89 (87.3) | 652 (97.9) | 1 | | | |
| Yes | 27 | 13 (12.7) | 14 (2.1) | 4.00 (2.58–6.20) | **P<.001** | 0.78 (0.47–1.31) | 0.362 |
| **Hepatitis B vaccination completion status** | | | | | | | |
| Incomplete | 747 | 84 (82.3) | 663 (99.6) | 1 | | | |
| Completed | 21 | 18 (17.7) | 3 (0.4) | 7.62 (5.83–9.95) | **P<.001** | 1.85 (1.26–2.70) | **0.001*** |
| **Attitude towards effectiveness of Hepatitis B vaccine** | | | | | | | |
| It is not effective | 587 | 35 (34.3) | 552 (82.9) | 1 | | | |
| It is effective | 181 | 67 (65.7) | 114 (17.1) | 6.20 (4.27–9.01) | **P<.001** | 3.67 (2.34–5.73) | **<.001*** |

* Considering a 95% CI, a p-value ≤ 0.05 was considered to be statistically significant in this study. CPR = Crude Prevalence Ratio, APR = Adjusted Prevalence Ratio

p = 0.016). Those who were unaware of the recommended vaccine doses for hepatitis B had a 57% lower prevalence of hepatitis B screening compared to those who were aware (aPR 0.43, 95% CI: 0.27–0.68, p<.001). Those who had completed the hepatitis B vaccine schedule had an 85% higher prevalence of hepatitis B screening compared to those who had not completed it (aPR 1.85, 95% CI: 1.26–2.70, p = 001). Young psychoactive substance users who believed the vaccine was effective against hepatitis B had a 134% higher prevalence of hepatitis B screening compared to those who felt it was ineffective (aPR 3.67, 95% CI: 2.34–5.73, p<.001).

## Hepatitis B vaccination status, knowledge and reasons for not being vaccinated

Only 44.4% (341/768) of the study participants had ever heard about hepatitis B vaccination. Only 8.4% (65/768) of the study participants knew the recommended vaccine dose (3 doses)

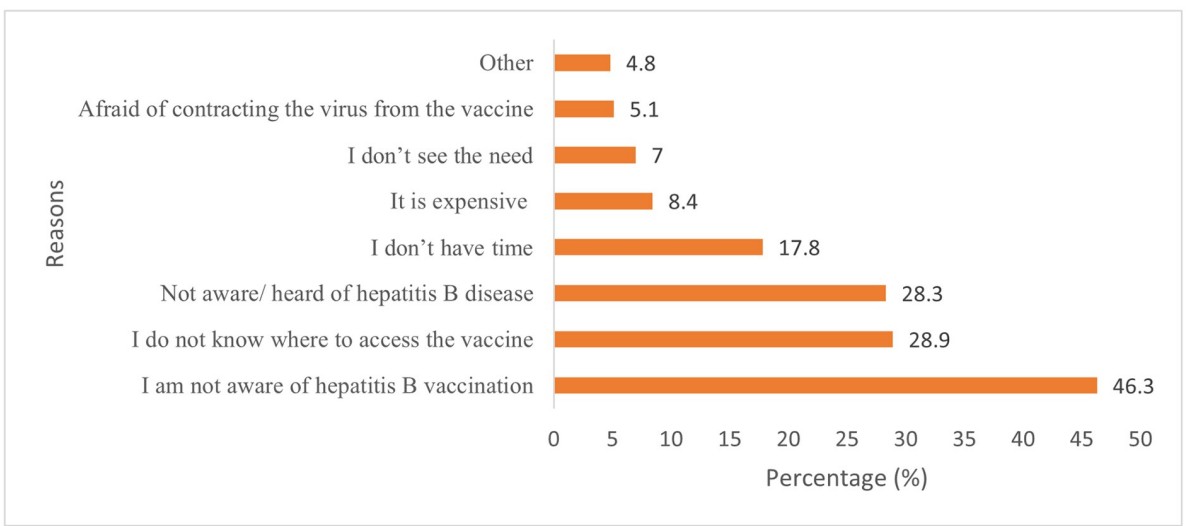

**Fig 3. Reasons young psychoactive substance users in Kampala's informal settlements gave for not being vaccinated against hepatitis B.**

for hepatitis B. Only 3.5% (27/768) of the study participants knew that the vaccine can protect them against hepatitis B for more than 25 years. Fig 3 shows the main reasons for not being vaccinated. About 46.3% (355/) of the study participants who had never received a hepatitis B vaccine dose mentioned that they were not aware of the vaccine; 28.9% (222/768) did not know where to access the vaccine; 28.3% (217/768) were not aware of the disease, and 8.4% (65/768) felt hepatitis vaccination was expensive (Fig 3).

## Predictors of completion of the hepatitis B vaccination schedule

Table 3 shows the predictors of completion of the hepatitis B vaccination schedule. The level of education and knowledge of the duration the vaccine was associated with hepatitis B vaccination at the bivariate level. At the multivariable level, knowledge of the recommended vaccine dose for hepatitis B, ever screening for hepatitis B and the belief that the vaccine is effective in preventing hepatitis B were statistically significantly associated with completion of the vaccination schedule. The odds of completing the hepatitis B vaccination schedule among young psychoactive substance users who did not know the recommended hepatitis B vaccine dose were 0.06 times lower compared to those who knew the recommended vaccine dose (aOR 0.06, 95% CI: 0.01–0.35). Whereas amongst young psychoactive substance users who had ever screened for hepatitis B, the odds of completing the hepatitis B vaccination schedule were 9.68 times higher compared to those who had never screened (aOR 9.68, 95% CI: 2.17–43.16, p = 0.003). The odds of hepatitis B vaccination schedule completion among young psychoactive substance users who felt the vaccine was effective in preventing hepatitis B were 11.8 times higher in comparison to those who felt it was not effective (aOR 11.80, 95% CI: 1.13–110.14, p = 0.039).

## Discussion

The current study aimed at establishing Hepatitis B screening and vaccination status of young psychoactive substance users in informal settlements in Kampala. We found low levels of HBV screening, HBV vaccination uptake, and low rates of completion of the vaccination schedule among this population. These findings are significant, given the current strategy of HBV

**Table 3. Predictors of the hepatitis B vaccination status of young psychoactive substance users in Kampala, Uganda.**

| Variable | Freq (n) | Vaccination status | | COR (95% CI) | P value | AOR (95% CI) | P value |
|---|---|---|---|---|---|---|---|
| | | Completed | Incomplete | | | | |
| **Sex of the respondents** | | | | | | | |
| Male | 603 | 13 (61.9) | 590 (79.0) | 1 | | | |
| Female | 165 | 8 (38.1) | 157 (21.0) | 2.31 (0.94–5.67) | 0.067 | 1.92 (0.50–7.35) | 0.337 |
| **Age category** | | | | | | | |
| 18–19 | 190 | 3 (14.3) | 187 (25.0) | 1 | | | |
| 20–24 | 578 | 18 (85.7) | 560 (75.0) | 2.00 (0.58–6.87) | 0.269 | 1.25 (0.23–6.63) | 0.793 |
| **Level of education** | | | | | | | |
| Primary | 322 | 4 (19.0) | 318 (42.6) | 1 | | | |
| Above primary | 446 | 17 (81.0) | 429 (57.4) | 3.15 (1.04–9.45) | **0.041** | 0.55 (0.10–2.80) | 0.474 |
| **Marital status** | | | | | | | |
| Single | 606 | 14 (66.7) | 592 (79.3) | 1 | | | |
| Married | 162 | 7 (33.3) | 155 (20.7) | 1.90 (0.75–4.81) | 0.17 | 2.49 (0.55–11.16) | 0.231 |
| **Living arrangements** | | | | | | | |
| Live with parents | 120 | 2 (9.5) | 118 (15.8) | 1 | | | |
| Independent | 648 | 19 990.5) | 629 (84.2) | 1.78 (0.40–7.75) | 0.441 | | |
| **Average monthly income** | | | | | | | |
| ≤ 68.0 | 496 | 17 (81.0) | 479 (64.1) | 1 | | | |
| 68.1–136 | 207 | 2 (9.5) | 205 (27.4) | 2.74 (0.62–1.20) | 0.086 | 0.22 (0.03–1.43) | 0.114 |
| Above 136 | 65 | 2 (9.5) | 63 (8.4) | 0.89 (0.20–3.96) | 0.883 | 0.50 (0.07–3.37) | 0.479 |
| **Know the recommended vaccine dose** | | | | | | | |
| Yes | 65 | 19 (90.5) | 46 (6.2) | 1 | | | |
| No | 703 | 2 (9.5) | 701 (93.8) | 0.01 (0.01–0.03) | **P<.001** | 0.06 (0.01–0.35) | **0.002**\* |
| **Know the duration the vaccine provides protection** | | | | | | | |
| No | 741 | 12 (57.1) | 729 (97.6) | 1 | | | |
| Yes | 27 | 9 (42.9) | 18 (2.4) | 30.37 (11.36–81.14) | **P<.001** | 3.62 (0.79–16.60) | 0.098 |
| **Ever screened for hepatitis B** | | | | | | | |
| No | 666 | 3 (14.3) | 663 (88.3) | 1 | | | |
| Yes | 102 | 18 (85.7) | 84 (11.2) | 47.3 (13.66–164.16) | P<.001 | 9.68 (2.17–43.16) | **0.003**\* |
| **Attitude towards effectiveness of Hep B vaccine** | | | | | | | |
| Not Effective | 587 | 1 (4.8) | 586 (78.5) | 1 | | 1 | |
| Effective | 181 | 20 (95.2) | 161 (21.5) | 72.7 (9.69–546.52) | **P<.001** | 11.8 (1.13–110.14) | **0.039**\* |

Considering a 95% CI, a p-value ≤ 0.05\* was considered to be statistically significant in this study. COR = Crude Odds Ratio, AOR = Adjusted Odds Ratio

micro-elimination by 2030 [41]. The strategy recommends the need to scale up hepatitis B prevention strategies to all underserved populations such as those residing in informal settlements. Screening for HBV infection is recommended for high-risk groups especially those with a prevalence of ≥2 [8]. However, only 13.3% of the young psychoactive substance users in this study had ever screened for hepatitis B despite a prevalence of 2.0%. This is mainly attributed to the lack of awareness of the HBV infection and associated preventive measures. Low awareness about HBV and its prevention has recently been reported in other key populations in Uganda [65]. Besides, available data indicate that informal settlements in Kampala are characterized by limited access to health care services [66], which could also have affected screening rates.

Hepatitis B screening was significantly associated with level of education, completion of the vaccination schedule and knowledge of the recommended vaccine dose. Females had a 61%

higher prevalence of hepatitis B screening compared to males. HBV screening may have been higher among females because it is recommended by the Ugandan MoH that every pregnant woman be screened for the disease during antenatal care so as to reduce vertical transmission [54]. Females are known to have better health seeking behaviours compared to males. This could have impacted on their hepatitis B screening rates. These findings are in agreement with those of Osei, Niyilapah [67] which indicated that females were more likely to screen for hepatitis B compared to males. In addition, males often show a reluctance in receiving health care services which could explain their low screening rates for hepatitis B [68, 69].

In some areas, especially urban settings, access to screening and vaccination services comes at a cost. This study showed that young psychoactive substance users with a higher level of income were more likely to screen for hepatitis B compared to those who had a lower level of income. This could be attributed to the high cost of accessing hepatitis B prevention services [70]. In some situations, health facilities providing these services are located further away from informal settlements. Therefore, young psychoactive substance users in informal settings incur transport costs to access hepatitis B prevention services. A lack of the financial resources therefore reduces the chances of low-income earners screening for HBV.

Completion of the hepatitis B vaccination schedule was also low, due to insufficient knowledge of the vaccine and HBV infection, and the fact that a significant proportion of young psychoactive substance users did not know where to access the vaccine. Limited access to health services has also been documented as a barrier to uptake of hepatitis B prevention services [18]. Our findings are also similar to a study among young injection drug users in the US where only 10% of younger participants reported having completed the hepatitis B vaccine series [71].

Young psychoactive substance users who had competed the hepatitis B vaccination schedule were more likely to have been screened. Usually, hepatitis B screening precedes vaccination in most healthcare facilities thus higher screening rates. Young psychoactive substance users who felt that the vaccine was effective in preventing hepatitis B viral infection were more likely to have screened for hepatitis B. This is because positive attitude has been shown to positively impact the uptake of prevention services.

Lack of knowledge of the recommended vaccine dose for hepatitis B was associated with a lesser likelihood of completing the hepatitis B vaccination schedule. Young psychoactive substance users who were aware of the recommended vaccine dose are likely to have been sensitised about hepatitis B. Being knowledgeable about hepatitis B may have impacted their attitude and health seeking behaviours. Consequently, they may have been motivated to take all the vaccine doses. A number of studies have concluded that an adequate level of knowledge of disease conditions increases the uptake of prevention services such as hepatitis B vaccination [67].

Young psychoactive substance users who had ever screened for hepatitis B were more likely to have completed the schedule than those who had never screened. Having screened for hepatitis B is an indicator of a better health seeking behaviour. In addition, those who had ever screened may have felt to be at an elevated risk of the HBV and therefore, undertaking the vaccination would protect them against the infection. Young psychoactive substance users who felt the vaccine was effective were more likely to have completed the vaccination schedule compared to those who did not. This re-echoes the fact that a positive attitude is more likely to positively impact preventive behaviours such as vaccination. Such individuals believe in the protective efficacy of the vaccine, and are bound to complete the vaccination schedule at all costs.

The strength of this study is that it provides useful insights into the predictors of screening and hepatitis B vaccination status of young psychoactive substance users in informal

settlements, an area that is less studied. However, there are some limitations. This study relied on self-reports that may be liable to social desirability bias for substance use and testing results. The cross-sectional design cannot establish causation between hepatitis B screening rates and completion of the hepatitis B vaccination schedule. This study did not apply sample weights, which could make it prone to unequal selection probability [72, 73]. Again, the study may have been affected by recall bias just like other vaccine uptake studies [74–76]. Lastly, these findings are only applicable to young psychoactive substance users in informal settlements and not young people in the general population.

## Conclusion and recommendations

This study indicates that both Hepatitis B screening and vaccination schedule completion rates are low among young psychoactive substance users. Generally, lack of knowledge and negative attitude towards the Hepatitis B vaccine were strong predictors of hepatitis B screening and completion of the Hepatitis B vaccination schedule. The predictors of Hepatitis B screening among young psychoactive substance users included sex, average monthly income, knowledge of the recommended Hepatitis B vaccine dose, Hepatitis B vaccination completion status and attitude towards effectiveness of the Hepatitis B vaccine. The predictors of completion of the Hepatitis B vaccination schedule among young psychoactive substance users in informal settlements included knowledge of the recommended vaccine dose, having ever screened for hepatitis B and attitude towards effectiveness of Hepatitis B vaccine. The findings by this study therefore, highlight the need for the Ministry of Health to spearhead the creation of awareness of young psychoactive substance users on the epidemiology of Hepatitis B. The Ministry of Health should also strengthen outreach programs on Hepatitis B with keen emphasis on high-risk subgroups such as those who use psychoactive substances.

## Supporting information

**S1 Dataset.**
(XLS)

## Acknowledgments

We would like to thank the study community for sparing their time to participate in this study. This study would not have been a success without the help of the community guides in the study informal settlements. Your effort in connecting us to the study participants is highly appreciated. Finally, thanks go out to all our research assistants (Nakiggala Joanna, Namulindwa Gloria, Soigi Christine, Nyakabwa Job, Andrew David Mugisha, Kiiza Ignatius and Bambuza Olivia) whose effort in undertaking the data collection is invaluable.

## Author Contributions

**Conceptualization:** Tonny Ssekamatte, John Bosco Isunju, Moses Tetui, Richard K. Mugambe, Esther Buregyeya, Simon P. S. Kibira.

**Formal analysis:** Tonny Ssekamatte, John Bosco Isunju, Joan Nankya Mutyoba, Moses Tetui, Chenai Kitchen, Simon P. S. Kibira.

**Methodology:** Tonny Ssekamatte, John Bosco Isunju, Joan Nankya Mutyoba, Moses Tetui, Richard K. Mugambe, Justine Nnakate Bukenya.

**Project administration:** Tonny Ssekamatte, Aisha Nalugya, Wagaba Brenda, Patience Oputan, Justine Nnakate Bukenya, Simon P. S. Kibira.

**Resources:** Tonny Ssekamatte, Richard K. Mugambe, Simon P. S. Kibira.

**Supervision:** Tonny Ssekamatte, Aisha Nalugya, Winnifred K. Kansiime, Wagaba Brenda, Patience Oputan, Simon P. S. Kibira.

**Validation:** Chenai Kitchen.

**Visualization:** Moses Tetui.

**Writing – original draft:** Tonny Ssekamatte, John Bosco Isunju, Joan Nankya Mutyoba, Moses Tetui, Richard K. Mugambe, Aisha Nalugya, Winnifred K. Kansiime, Chenai Kitchen, Wagaba Brenda, Patience Oputan, Justine Nnakate Bukenya, Esther Buregyeya, Simon P. S. Kibira.

**Writing – review & editing:** Tonny Ssekamatte, John Bosco Isunju, Joan Nankya Mutyoba, Moses Tetui, Richard K. Mugambe, Aisha Nalugya, Winnifred K. Kansiime, Chenai Kitchen, Wagaba Brenda, Patience Oputan, Justine Nnakate Bukenya, Esther Buregyeya, Simon P. S. Kibira.

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
