## [Decision Letter · Decision Letter 0]

24 Sep 2021

PONE-D-21-18903Hepatitis B screening and vaccination status of young psychoactive substance users in informal settlements in Kampala, UgandaPLOS ONE

Dear Dr. Ssekamatte,

Thank you for submitting your manuscript to PLOS ONE. After careful consideration, we feel that it has merit but does not fully meet PLOS ONE’s publication criteria as it currently stands. Therefore, we invite you to submit a revised version of the manuscript that addresses the points raised during the review process. See the peer-review comments attached. 

We look forward to receiving your revised manuscript.

Kind regards,

Hamidreza Karimi-Sari, MD

Academic Editor

PLOS ONE

Journal Requirements:

Reviewers' comments:

Reviewer's Responses to Questions

**Comments to the Author**

1. Is the manuscript technically sound, and do the data support the conclusions?

Reviewer #1: Yes

Reviewer #2: Yes

2. Has the statistical analysis been performed appropriately and rigorously? 

Reviewer #1: Yes

Reviewer #2: Yes

3. Have the authors made all data underlying the findings in their manuscript fully available?

Reviewer #1: Yes

Reviewer #2: No

4. Is the manuscript presented in an intelligible fashion and written in standard English?

Reviewer #1: Yes

Reviewer #2: Yes

5. Review Comments to the Author

Reviewer #1: The present manuscript is a well-written cross-sectional study on predictors of hepatitis B vaccination and screening among young adults with a history of psychoactive substance use. Authors have found some significant associations for socioeconomic indicators and lack of knowledge. In general, the manuscript can be considered for publication after minor revisions.

Background: This section can be shortened to 3-4 structured paragraphs in order to improve the flow of the background. There is some unnecessary information that can be removed, especially in paragraph 3, and the 4th and 5th paragraphs can be summarized and combined into one paragraph. Also, the first sentence of background should be grammatically revised.

Methods: There is no need to name all divisions and hospitals in Kampala here. The last sentence of scope and design and the second sentence of study population belongs to the background. What is Khat? Please define.

Results: Some figures and tables are located incorrectly in the manuscript. Please check.

For example, the section “predictors of hepatitis B screening” does not represent the data of figure 2, while the section “Hepatitis B vaccination status” is describing figure 2. Also, table 2, 3, and 4 has been wrongly named in the text. This sentence should be revised: “only 2.7% (21/768) had received two vaccine doses while only 2.7% (21/768) had completed the hepatitis B vaccination schedule of 3 doses.”

In addition to the above, there are some minor grammatical and linguistic errors that needs proofreading.

Reviewer #2: Unable to locate where the data is available without restriction.

Please explain what is meant by the term “settlement”. Is it permanent housing? Temporary? How long do people stay and why? Do they live in groups? Families? Give a rationale for excluding those who have lived there <6 months.

Revision should address vaccine related policies. If vaccines are required/recommended in childhood, recall bias is a major concern and should be described in the limitations. Be certain to reference other studies related to vaccine recall.

No reference is provided in the introduction for the statement “they are more likely to be unaware of their Hepatitis B status, and unvaccinated”. If there is no evidence to support it, delete the statement.

Are use of marijuana and alcohol associated with risk of hepatitis B? Please spend some time explaining the theoretical framework to help readers understand why you included all substance users and not just injection drug users. Explain how multiple risk factors tend to co-occur, etc.

The theoretical framework should also address how covariates were chosen. It’s not appropriate to let the model decide which variables to include- please explain the rationale and revise your models appropriately. Consider causal pathways and do not adjust for factors that lie in the pathway.

How many were excluded due to intoxication? Clarify how this is accounted for in the response rate.

What time of day was data collection performed?

Please spell out what you mean by RDS and fully explain how potential respondents were approached. Tone down the use of the word “always” unless you have evidence to back up such a strong claim.

Remove any language related to unweighted analyses from your description of the methods. This makes it sound like you have weighted data and chose not to use the weights. This is misleading for readers. Given the study population, it would not be possible/practical to weight the data, and this is the only rationale- do not site a statistical rationale.

Are females more likely to be screened because of pregnancy-related practices? Are pregnant women routinely screened for hepatitis B?

Remove the word “variable” from figure 2.

What is meant by “sensitization”?

6. PLOS authors have the option to publish the peer review history of their article (what does this mean?). If published, this will include your full peer review and any attached files.

Reviewer #1: **Yes: **Sanam Hariri

Reviewer #2: No

---

## [Author Response · Author response to Decision Letter 0]

1 Mar 2022

Response to comments

Reviewer #1

1. Comment: The present manuscript is a well-written cross-sectional study on predictors of hepatitis B vaccination and screening among young adults with a history of psychoactive substance use. Authors have found some significant associations for socioeconomic indicators and lack of knowledge. In general, the manuscript can be considered for publication after minor revisions.

Response: Thank you for appreciating our efforts. We have revised the manuscript as recommended 

2. Comment: Background: This section can be shortened to 3-4 structured paragraphs in order to improve the flow of the background. There is some unnecessary information that can be removed, especially in paragraph 3, and the 4th and 5th paragraphs can be summarized and combined into one paragraph. Also, the first sentence of background should be grammatically revised.

Response: Thank you. The section has been revised accordingly

3. Comment: Methods: There is no need to name all divisions and hospitals in Kampala here. The last sentence of scope and design and the second sentence of study population belongs to the background. 

Response: Thank you. This paragraph has been revised (Page 6). 

4. Comment: What is Khat? Please define.

Response: Khat has been defined (page 6, lines 148-149)

5. Comment: Results: Some figures and tables are located incorrectly in the manuscript. Please check. For example, the section “predictors of hepatitis B screening” does not represent the data of figure 2, while the section “Hepatitis B vaccination status” is describing figure 2. Also, table 2, 3, and 4 has been wrongly named in the text. 

Response: Thank you. These have been changed (Page 12, line 271)

6. Comment: This sentence should be revised: “only 2.7% (21/768) had received two vaccine doses while only 2.7% (21/768) had completed the hepatitis B vaccination schedule of 3 doses.”

Response: Thank you. The sentence has been revised. Page 12 lines 277-278

7. Comment: In addition to the above, there are some minor grammatical and linguistic errors that needs proofreading.

Response: Thank you. We have proofread the paper and addressed the grammatical and linguistic errors.

Reviewer #2

1. Comment: Unable to locate where the data is available without restriction.

Response: The data have been uploaded as a supplementary file

2. Comment: Please explain what is meant by the term “settlement”. Is it permanent housing? Temporary? How long do people stay and why? Do they live in groups? Families? 

Response: The term settlement has been defined n the methods section. Page 6 lines 142-144277-27

3. Give a rationale for excluding those who have lived there <6 months.

Response: The rationale has been added. Since we used respondent-driven sampling, it would have been difficult for the peers to ascertain whether the prospective respondents actually lived in the selected informal settlements and or if they were actually substance users if they had lived there for less than 6 months. 

4. Comment: Revision should address vaccine related policies. If vaccines are required/recommended in childhood, recall bias is a major concern and should be described in the limitations. Be certain to reference other studies related to vaccine recall.

Response: It is less likely that recall bias would be an issue since all the respondents were born before hepatitis B was integrated into the national immunization programme (2002) which requires newborns to receive a short at 6, 10 and 14 weeks. Nonetheless we acknowledge that recall bias could still be a challenge and has been included n the study limitations. Page 20 Lines 411-412

5. Comment: No reference is provided in the introduction for the statement “they are more likely to be unaware of their Hepatitis B status, and unvaccinated”. If there is no evidence to support it, delete the statement

Response: Reference has been provided for that statement. Page 5 lines 11-119

6. Comment: Are use of marijuana and alcohol associated with risk of hepatitis B? Please spend some time explaining the theoretical framework to help readers understand why you included all substance users and not just injection drug users. 

Response: This has been explained on Page 6 Lines 151-153

7. Explain how multiple risk factors tend to co-occur, etc.

The theoretical framework should also address how covariates were chosen. It’s not appropriate to let the model decide which variables to include- please explain the rationale and revise your models appropriately. Consider causal pathways and do not adjust for factors that lie in the pathway.

Response: The selection of covariates has been addressed. Page Lines 191-202. The models have been revised accordingly. Page 17 and Page 14

8. Comment: How many were excluded due to intoxication? Clarify how this is accounted for in the response rate.

Response: Given the nature of recruitment we dd not account for non-response. Prior to recruitment we explained to both the primary and secondary seeds the criteria for inclusion in the study. This reduced the chances of recruiting individuals who were not eligible to zero. This was further confirmed with the screening tool. Page 7 Lines 179-181 

9. What time of day was data collection performed?

Response: Data were collected between 8:00 am and 6:00 pm throughout the data collection period. This has been included in the manuscript.

10. Comment: Please spell out what you mean by RDS and fully explain how potential respondents were approached. Tone down the use of the word “always” unless you have evidence to back up such a strong claim.

Response: RDS has been spelt out. The way potential respondents were approached s reported in our earlier publications. A reference has been given . Page 7 Lines 177-17

We have toned down the use of the word “always”

11. Comment: Remove any language related to unweighted analyses from your description of the methods. This makes it sound like you have weighted data and chose not to use the weights. This is misleading for readers. Given the study population, it would not be possible/practical to weight the data, and this is the only rationale- do not site a statistical rationale.

Response: All language related to unweighted analyses has been removed.

12. Comment: Are females more likely to be screened because of pregnancy-related practices? Are pregnant women routinely screened for hepatitis B?

Response: True. Although it is not yet part of the standard of care, women are screened for HBV during pregnancy. This has been incorporated in the discussion. Page 18 Page Lines 363-365

13. Remove the word “variable” from figure 2.

Response: The word “variable” has been removed from figure 2.

14. What is meant by “sensitization”?

Response: This has been replaced with “creating awareness.” Page 20 Line 421

---

## [Decision Letter · Decision Letter 1]

14 Mar 2022

PONE-D-21-18903R1Hepatitis B screening and vaccination status of young psychoactive substance users in informal settlements in Kampala, UgandaPLOS ONE

Dear Dr. Ssekamatte,

Thank you for submitting your manuscript to PLOS ONE. After careful consideration, we feel that it has merit but does not fully meet PLOS ONE’s publication criteria as it currently stands. Therefore, we invite you to submit a revised version of the manuscript that addresses the points raised during the review process.

We look forward to receiving your revised manuscript.

Kind regards,

Hamidreza Karimi-Sari, MD

Academic Editor

PLOS ONE

Journal Requirements:

Review Comments to the Author

Reviewer #2: 

No further comments, thank you for diligently addressing concerns. This is important research and you should be proud to see it in print.

Reviewer #3: 

Authors have tried to answer all comments from the reviewers and the manuscript is much improved, particularly the methods and results sections. Here are some additional minor comments:

The abstract contains many repetitive words and needs to be rewritten to sound good. There is no need to repeat the title in the abstract.

As I mentioned before, the introduction is long and contain some unnecessary or unrelated information. Authors may try to be concise and focused on the main problem that the current study is going to address. Also, this section can be shortened into 4-5 paragraphs.

In general, discussion is well-written but can also be more summarized and some sentences can be combined. There are terms like young psychoactive substance users or informal settlements in Kampala that repeats many times throughout the manuscript. Avoid repetitions like this: …vaccination status of young psychoactive substance users in informal settlements in Kampala. Among young psychoactive substance users living in urban informal settlements, we found... (Discussion-1st paragraph)

---

## [Author Response · Author response to Decision Letter 1]

16 Apr 2022

Response to comments

Journal Requirements:

Response: The comments have been checked for completeness and correctness 

Reviewer #3: 

1. Comment: Authors have tried to answer all comments from the reviewers and the manuscript is much improved, particularly the methods and results sections. Here are some additional minor comments:

Response: Thank you for acknowledging our efforts in responding to the comments. The minor comments are also appreciated.

2. The abstract contains many repetitive words and needs to be rewritten to sound good. There is no need to repeat the title in the abstract.

Response: The abstract has been revised to reduce repetitions. Pages 2-3 lines 40-71

3. As I mentioned before, the introduction is long and contain some unnecessary or unrelated information. Authors may try to be concise and focused on the main problem that the current study is going to address. Also, this section can be shortened into 4-5 paragraphs.

Response: Thank you. The introduction has been shortened to 5 paragraphs. Page 4-5 lines 74-118

4. In general, discussion is well-written but can also be more summarized and some sentences can be combined. There are terms like young psychoactive substance users or informal settlements in Kampala that repeats many times throughout the manuscript. Avoid repetitions like this: …vaccination status of young psychoactive substance users in informal settlements in Kampala. Among young psychoactive substance users living in urban informal settlements, we found... (Discussion-1st paragraph)

Response: The discussion has been shortened. Repetitions have been removed from the discussion and other parts of the manuscript. Page 17 lines 334-403

---

## [Editor Report · Decision Letter 2]

20 Apr 2022

Predictors of Hepatitis B screening and vaccination status of young psychoactive substance users in informal settlements in Kampala, Uganda

PONE-D-21-18903R2

Dear Dr. Ssekamatte,

We’re pleased to inform you that your manuscript has been judged scientifically suitable for publication and will be formally accepted for publication once it meets all outstanding technical requirements.

Kind regards,

Hamidreza Karimi-Sari, MD

Academic Editor

PLOS ONE

---

## [Editor Report · Acceptance letter]

27 Apr 2022

PONE-D-21-18903R2 

 Predictors of Hepatitis B screening and vaccination status of young psychoactive substance users in informal settlements in Kampala, Uganda 

Dear Dr. Ssekamatte:

I'm pleased to inform you that your manuscript has been deemed suitable for publication in PLOS ONE. Congratulations! Your manuscript is now with our production department. 

Kind regards, 

on behalf of

Dr. Hamidreza Karimi-Sari 

Academic Editor

PLOS ONE